# Unleashing the Power of One-Step Diffusion based Image Super-Resolution via a Large-Scale Diffusion Discriminator

**Jianze Li**[1*], **Jiezhang Cao**[2*], **Zichen Zou**[1], **Xiongfei Su**[3], **Xin Yuan**[4], **Yulun Zhang**[1†],
**Yong Guo**[4], **Xiaokang Yang**[1]

[1]Shanghai Jiao Tong University,  [2]Harvard University,  [3]China Mobile Research Institute,
[4]Westlake University,  [4]South China University of Technology

## Abstract

Diffusion models have demonstrated excellent performance for real-world image super-resolution (Real-ISR), albeit at high computational costs. Most existing methods are trying to derive one-step diffusion models from multi-step counterparts through knowledge distillation (KD) or variational score distillation (VSD). However, these methods are limited by the capabilities of the teacher model, especially if the teacher model itself is not sufficiently strong. To tackle these issues, we propose a new One-Step **D**iffusion model with a larger-scale **D**iffusion **D**iscriminator for SR, called $D^3SR$. Our discriminator is able to distill noisy features from any time step of diffusion models in the latent space. In this way, our diffusion discriminator breaks through the potential limitations imposed by the presence of a teacher model. Additionally, we improve the perceptual loss with edge-aware DISTS (EA-DISTS) to enhance the model's ability to generate fine details. Our experiments demonstrate that, compared with previous diffusion-based methods requiring dozens or even hundreds of steps, our $D^3SR$ attains comparable or even superior results in both quantitative metrics and qualitative evaluations. Moreover, compared with other methods, $D^3SR$ achieves at least $3\times$ faster inference speed and reduces parameters by at least 30%. We will release code and models at https://github.com/JianzeLi-114/D3SR.

## 1 Introduction

Real-world image super-resolution (Real-ISR) is a challenging task that aims to reconstruct high-resolution (HR) images from their low-resolution (LR) counterparts in real-world settings [1]. Most image super-resolution (SR) methods [2, 3, 4, 5, 6] use Bicubic downsampling of HR images to generate LR samples. These methods achieve good results in reconstructing simple degraded images. However, they struggle with the complex and unknown degradations widely existing in real-world scenarios. Previous research has predominantly employed generative adversarial networks (GANs) [7, 8, 9, 10] for Real-ISR task. In recent developments, diffusion models (DMs) [11, 12, 13] have emerged as a promising alternative. Recent Real-ISR methods have achieved outstanding performance by using diffusion models. In particular, some methods [14, 15, 16, 17, 18] leverage powerful pre-trained diffusion models, such as large-scale text-to-image (T2I) diffusion models like Stable Diffusion [12, 19]. These pre-trained T2I models provide extensive priors and powerful generative abilities. Most Diffusion model based methods generate HR images by employing ControlNet models [20], conditioning on the LR inputs. However, these methods typically require

---

*Equal contribution
†Corresponding author: Yulun Zhang, yulun100@gmail.com

tens to hundreds of diffusion steps to produce high-quality HR images. The introduction of ControlNet not only increases the number of model parameters but also further exacerbates inference latency. Consequently, Diffusion model based multi-step diffusion methods often incur delays of tens of seconds when processing a single image, which significantly limits their practical application in real-world scenarios for low-level image reconstruction tasks.

To accelerate the generation process of diffusion models, recent research has introduced numerous one-step diffusion methods [21, 22, 23, 24, 25, 26, 27, 28]. These methods are known as diffusion distillation, which distill multi-step pre-trained diffusion models into one-step counterparts. Most of these approaches employ a knowledge distillation strategy, using the multi-step diffusion model as a teacher to train a one-step diffusion student model. These methods significantly reduce inference latency, and the quality of the generated images can be comparable to that of multi-step diffusion models. Real-ISR methods based on one-step diffusion models have become an increasingly popular research direction [29, 30, 31, 32, 33, 34]. These methods employ pre-trained multi-step diffusion models as teachers to guide the training process. They achieve promising results, with performance comparable to that of multi-step models. However, further performance improvement remains a challenging task. Existing methods rely on a multi-step teacher to conduct distillation. Nevertheless, this paradigm would inevitably come with several limitations. **First**, the multi-step teacher may hamper the effectiveness of distillation if the teacher itself is not strong enough. For example, variational score distillation (VSD) [35, 36, 32] is a one-step diffusion distillation method. It enhances the realism of generated images by optimizing the KL divergence between the scores of the teacher and student models. This approach heavily relies on the prior knowledge of the teacher diffusion model. The limitations of the multi-step diffusion model impose an upper bound on the performance of the aforementioned methods. **Second**, the widely used VSD approach does not exploit any high resolution (HR) data for training but merely depend on the prior knowledge embedded in the pre-trained parameters. If the distribution fitted by the multi-step diffusion model (teacher in VSD) deviates from the high-quality image distribution, it may lead to a loss of realism or the generation of fake textures in the student model's images.

To overcome the aforementioned challenges, we propose a novel one-step diffusion model with a large-scale diffusion discriminator. Different from existing one-step diffusion models using distillation techniques with teacher models, we use a larger-scale diffusion model, SDXL [19], as a discriminator to leverage the powerful priors. Specifically, our diffusion discriminator aims to distill latent features with different noises from true data. At the same time, it enables us to perceive the true data distribution of high-quality super-resolution datasets. As a result, we overcome the performance limitations imposed by the teacher model's upper bound. Additionally, we propose a simple and effective improvement to the perceptual loss, edge-aware DISTS (EA-DISTS), by capturing high-frequency details from extracted edges. Our comprehensive experiments indicate that $D^3$SR achieves superior performance and less inference time among one-step DM-based Real-ISR models.

Our main contributions are summarized as follows:

- We propose a simple but effective one-step diffusion distillation method, $D^3$SR, which uses a large-scale diffusion model as a discriminator for adversarial training. Unlike previous one-step diffusion ISR methods, we do not use a pre-trained multi-step diffusion model as a teacher to guide the training. Instead, we employ a larger-scale diffusion model to guide the training process. This breaks through the performance limitations of teacher models in previous distillation methods.

- We improve the perceptual loss by proposing the edge-aware DISTS (EA-DISTS) loss. Our EA-DISTS leverages image edges to enhance the model's ability and improve the authenticity of reconstructed details.

- Experiments show that our method outperforms previous one-step diffusion distillation methods, such as VSD and knowledge distillation. When compared with multi-step DM-based models, $D^3$SR obtains comparable or even better performance with over $7\times$ speedup in inference time. Moreover, our method offers a $3\times$ inference speed advantage over one-step DM-based methods and reduces parameters by at least 30%.

## 2 Related Work

### 2.1 Real-World Image Super-Resolution

Real-world image super-resolution (Real-ISR) aims to recover high-resolution (HR) images from low-resolution (LR) observations in real-world scenarios. The complex and unknown degradation patterns in such scenarios make Real-ISR a challenging problem [37, 38, 39, 40]. To address this problem, a variety of methods have been proposed. Early image super-resolution models [2, 41, 42, 5, 6] typically rely on simple synthetic degradations like Bicubic downsampling for generating LR-HR pairs. Although these methods perform well under simple degradation settings, they struggle to achieve satisfactory results in real-world scenarios. Later, GAN-based methods such as BSRGAN [9], Real-ESRGAN [8], and SwinIR-GAN [10] introduce more complex degradation processes. These methods achieve promising perceptual quality but encounter issues such as training instability. Additionally, they have limitations in preserving fine natural details. Recently, Stable Diffusion (SD) [12] is considered for addressing Real-ISR tasks due to its strong ability to capture complex data distributions and provide robust generative priors. Approaches such as StableSR [43], DiffBIR [16], and SeeSR [14] leverage pre-trained diffusion priors and ControlNet models [20] to enhance HR image generation. One-step diffusion models have gained widespread attention from researchers. Methods such as YONOS-SR [29], SinSR [30], OESEDiff [32], AddSR [31], TAD-SR [44], and TSD-SR [45] have achieved Real-ISR with diffusion models in a single sampling step.

### 2.2 Acceleration of Diffusion Models

Acceleration of diffusion models can reduce computational costs and inference time. Therefore, various strategies have been developed to enhance the efficiency of diffusion models in image generation tasks. Fast diffusion samplers [13, 46, 47, 48, 49, 50] have significantly reduced the number of sampling steps from 1,000 to 15~100 without requiring model retraining. However, further reducing the steps below 10 often leads to a performance drop. Under these circumstances, distillation techniques have made considerable progress in speeding up inference [21, 51, 23, 24, 25, 52, 26, 27, 53, 36]. For instance, Progressive Distillation (PD) methods [23, 24] have distilled pre-trained diffusion models to under 10 steps. Consistency models [25] have further reduced the steps to 2~4 with promising results. Instaflow [27] further achieves one-step generation through reflow [51] and distillation. Recent score distillation-based methods, such as Distribution Matching Distillation (DMD) [54, 55] and Variational Score Distillation (VSD) [56, 36], aim to achieve one-step text-to-image generation. They minimize the Kullback–Leibler (KL) divergence between the generated data distribution and the real data distribution. Although these approaches have made notable progress, they still face challenges, like high training costs and dependence on teacher models.

## 3 Method

In this section, we present our real-world image super-resolution (Real-ISR) model D$^3$SR. First, in section 3.1, we review the basics of diffusion models and introduce the D$^3$SR generator. In section 3.2, we introduce the diffusion distillation method using a large-scale diffusion discriminator. In section 3.3, we introduce the edge-aware DISTS (EA-DISTS) perceptual loss. This loss improves texture details and enhances visual quality. Finally, in section 3.4, we describe the training process for D$^3$SR.

### 3.1 Preliminaries: Diffusion Models

Diffusion models include forward and reverse processes. During the forward diffusion process, Gaussian noise with variance $\beta_t \in (0, 1)$ is gradually injected into the latent variable $z$: $z_t = \sqrt{\bar{\alpha}_t} z + \sqrt{1 - \bar{\alpha}_t} \epsilon$, where $\epsilon \sim \mathcal{N}(0, \mathbf{I})$, $\alpha_t = 1 - \beta_t$, and $\bar{\alpha}_t = \prod_{s=1}^{t} \alpha_s$. In the reverse process, we can directly predict the clean latent variable $\hat{z}_0$ from the model's predicted noise $\hat{\epsilon}$: $\hat{z}_0 = \frac{z_t - \sqrt{1 - \bar{\alpha}_t} \hat{\epsilon}}{\sqrt{\bar{\alpha}_t}}$, where $\hat{\epsilon}$ is the prediction of the network $\epsilon_\theta$ given $z_t$ and $t$: $\hat{\epsilon} = \epsilon_\theta(z_t; t)$.

As illustrated in Fig. 1, we first employ the encoder $E_\theta$ to map the low-resolution (LR) image $x_L$ into the latent space, yielding $z_L$: $z_L = E_\theta(x_L)$. Next, we perform a one denoising step to obtain the predicted noise $\hat{\epsilon}$ and compute the high-resolution (HR) latent representation $\hat{z}_H$:

$$\hat{z}_H = \frac{z_L - \sqrt{1 - \bar{\alpha}_{T_L}} \, \epsilon_\theta(z_L; T_L)}{\sqrt{\bar{\alpha}_{T_L}}}, \tag{1}$$

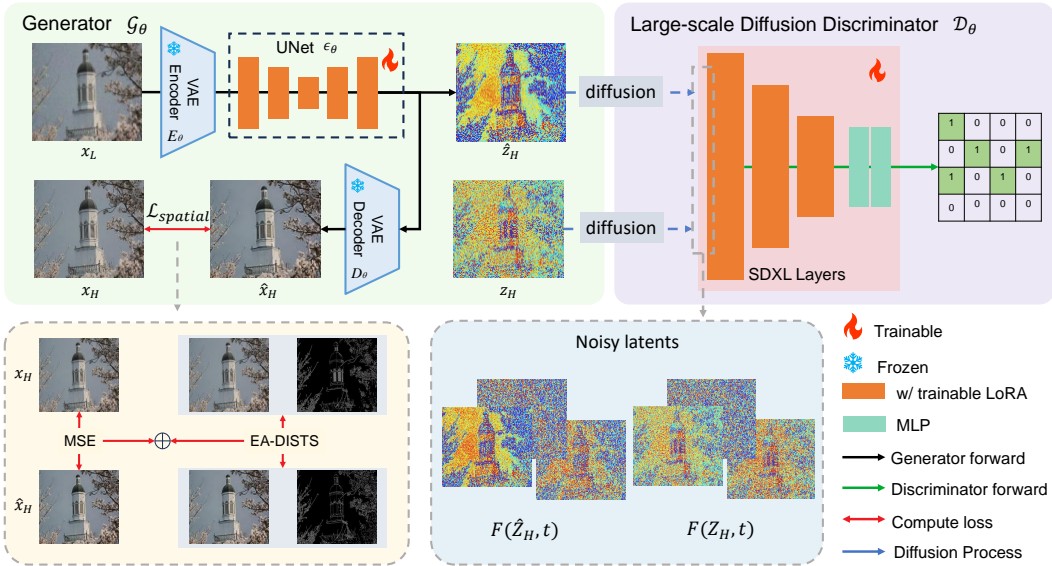

Figure 1: Training framework of D³SR. The left side represents the generator $\mathcal{G}_\theta$, which includes the pre-trained VAE and UNet from Stable Diffusion. Only the UNet is fine-tuned using LoRA, while other parameters remain frozen. The right side depicts the diffusion discriminator , which guides the training process without participating in inference. The discriminator extracts the UNet Mid-block outputs and processes them through an MLP to generate realism scores for different image regions. Both the downsample and middle blocks of the UNet in the discriminator are fine-tuned with LoRA, whereas the MLP is randomly initialized.

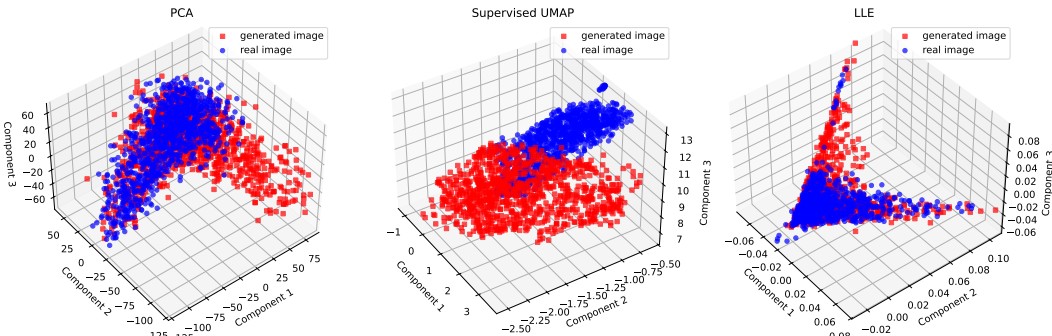

Figure 2: Visualization of features dimensionality reduction for the first 100 channels from the middle block outputs of the Stable Diffusion (SD) UNet. The distributions of the two types of image features are distinctly different.

where $\epsilon_\theta$ denotes the denoising network parameterized by $\theta$, and $T_L$ is the diffusion time step. Unlike one-step text-to-image (T2I) diffusion models [25, 54], the input to the UNet of the Real-ISR diffusion models is not pure Gaussian noise. We set $T_L$ to an intermediate time step within the range $[0, T]$, where $T$ is the total number of diffusion time steps. In Stable Diffusion (SD), $T = 1,000$. Finally, we decode $\hat{z}_H$ using the decoder $D_\theta$ to reconstruct the HR image $\hat{x}_H$: $\hat{x}_H = D_\theta(\hat{z}_H)$. The entire computation process of the generator can be expressed as $\hat{x}_H = \mathcal{G}_\theta(x_L)$.

## 3.2 Distillation with Large Diffusion Discriminator

Currently successful one-step diffusion distillation methods applied in image super-resolution include knowledge distillation (KD), variational score distillation (VSD), and others. Among them, VSD stands out due to its interesting principles and excellent performance. VSD works by training an additional diffusion network to fit the fake score $s_{\text{fake}}$ of the generated image, while using a frozen pre-trained diffusion model to obtain the real score $s_{\text{real}}$ based on its prior. The method then aligns the two scores' differences using the Kullback-Leibler (KL) divergence, thus enhancing the realism

of the generated image. However, in VSD, the absence of real image datasets implies that the upper bound of VSD is limited by the prior of the pre-trained diffusion model. The teacher model in VSD restricts the generative capacity of the student model, making further performance improvements challenging.

To address the issue incurred by a weak multi-step teacher, we propose to use a large-scale diffusion model to provide stronger guidance for one-step distillation. Figure 2 shows the distributions of real and generated images' latent code at the middle block output of the UNet in the Stable Diffusion (SD) model. There is a clear difference in their distribution patterns. This suggests that using a pre-trained diffusion model as a discriminator has great potential. It can also avoid instability issues during the early stages of training. Figure 1 illustrates our training framework. We append an MLP block after the SD UNet middle block as a classifier to output the authenticity score for each patch.

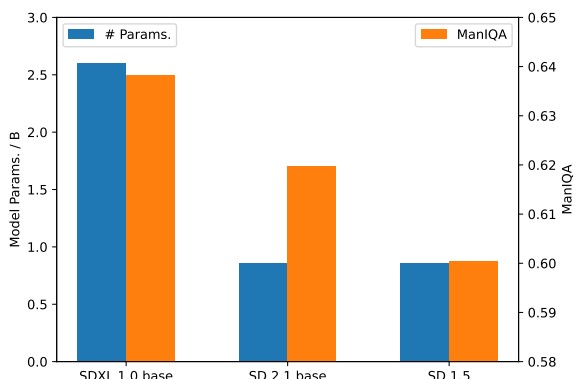

Figure 3: Comparison of the performance of SD models with different scales as discriminators. As the model size increases, the performance of the generator improves accordingly.

During training, we input the forward diffusion results of both the generator's predicted latent code $\hat{z}_H$, and the ground truth latent code $z_H = E_\theta(x_H)$. The adversarial losses for updating the generator and discriminator are defined as:

$$\mathcal{L}_\mathcal{G} = -\mathbb{E}_{x_L \sim p_{\text{data}}, \, t \sim [0,T]} \left[ \log \mathcal{D}_\theta \left( F \left( \hat{z}_H, t \right) \right) \right], \tag{2}$$

$$\mathcal{L}_\mathcal{D} = -\mathbb{E}_{x_L \sim p_{\text{data}}, \, t \sim [0,T]} \left[ \log \left( 1 - \mathcal{D}_\theta \left( F \left( \hat{z}_H, t \right) \right) \right) \right]$$
$$- \mathbb{E}_{x_H \sim p_{\text{data}}, \, t \sim [0,T]} \left[ \log \mathcal{D}_\theta \left( F \left( z_H, t \right) \right) \right], \tag{3}$$

where $F(\cdot, t)$ denotes the forward diffusion process of $\cdot$ at time step $t \in [0, T]$, specifically,

$$F(z, t) = \sqrt{\bar{\alpha}_t}\, z + \sqrt{1 - \bar{\alpha}_t}\, \epsilon, \quad \text{with } \epsilon \sim \mathcal{N}(0, \mathbf{I}). \tag{4}$$

**Relation to Diffusion GAN Methods.** Recently, many methods have combined GANs and Diffusion models, achieving success in image generation and other fields. Diffusion-GAN [57] uses a timestep-dependent discriminator to guide the generator's training. DDGAN [58] employs a multimodal conditional GAN to achieve large-step denoising. ADD [22] and LADD [59] both introduce discriminators to enhance the generative quality of one-step student diffusion models. These discriminators operate in pixel space and latent space, respectively. Both our method and existing Diffusion GAN methods demonstrate the potential of adversarial training in diffusion models. However, there are several essential differences. **First**, we use a pre-trained multi-step diffusion model as the discriminator, leveraging the prior of large-scale models to guide the training process. **Second**, we explore the impact of scaling. Figure 3 shows the performance of different SD models as discriminators. It reveals that large-scale diffusion discriminators break through the performance limits of pre-trained diffusion models, achieving superior results.

### 3.3 Edge-Aware DISTS

To further enhance the quality of the generated images, we aim to incorporate perceptual loss. Most image reconstruction methods utilize LPIPS [60] as the perceptual loss. However, to better preserve image texture details and alleviate pseudo-textures in the reconstruction under higher noise levels, we need to focus on the textures on HR images. DISTS [61] can compute the structural and textural similarity of images, aligning with human subjective perception of image quality. Furthermore, regions with rich textures or details often exhibit strong edge information. Leveraging image edge information effectively enhances texture quality. Based on this, we propose a novel perceptual loss, termed Edge-Aware DISTS (EA-DISTS). This perceptual loss simultaneously evaluates the structure

and texture similarity of the reconstructed and HR images and their edges, thereby enhancing texture detail restoration.

Our proposed EA-DISTS is defined as:

$$
\begin{aligned}
\mathcal{L}_{\text{EA-DISTS}}(\mathcal{G}_\theta(x_L), x_H) = \\
\mathcal{L}_{\text{DISTS}}(\mathcal{G}_\theta(x_L), x_H) + \mathcal{L}_{\text{DISTS}}(\mathcal{S}(\mathcal{G}_\theta(x_L)), \mathcal{S}(x_H)),
\end{aligned}
\tag{5}
$$

where $\mathcal{S}(\cdot)$ represents the Sobel operator used to extract edge information from the images. It consists of two convolution kernels, $G_x$ and $G_y$, which detect horizontal and vertical edges, respectively:

$$
G_x = \begin{bmatrix} -1 & 0 & 1 \\ -2 & 0 & 2 \\ -1 & 0 & 1 \end{bmatrix}, \quad G_y = \begin{bmatrix} -1 & -2 & -1 \\ 0 & 0 & 0 \\ 1 & 2 & 1 \end{bmatrix}.
\tag{6}
$$

The Sobel operator is applied to an image $x$ as follows:

$$
\mathcal{S}(x) = \sqrt{(G_x * x)^2 + (G_y * x)^2},
\tag{7}
$$

where $*$ denotes the convolution operation.

To intuitively demonstrate the effectiveness of EA-DISTS, we visualize the feature maps during the DISTS computation process. Figure 4 presents the visualization results of VGG-16 feature maps. As shown in Fig. 4, in areas rich with image details, such as the building windows, the feature maps associated with EA-DISTS exhibit more high-frequency information. Compared to DISTS, EA-DISTS demonstrates higher contrast in textured and smooth regions, further emphasizing the textural details within the images. Our EA-DISTS

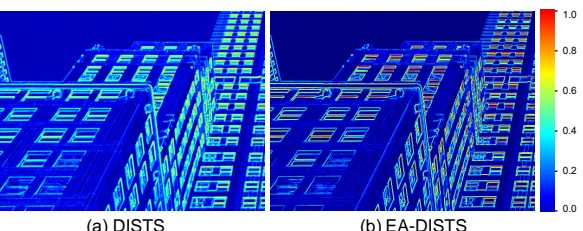

(a) DISTS    (b) EA-DISTS

Figure 4: Feature visualization associated with DISTS and EA-DISTS. Our EA-DISTS captures more high-frequency information, like texture and edges.

places greater emphasis on texture details within images, guiding the model to generate realistic and rich details.

### 3.4 Overall Training Scheme

Here, we summarize the whole one-step diffusion model training process. As described in section 3.1, within the generator component, D$^3$SR obtains $\hat{z}_H$ and the decoded high-resolution image $\hat{x}_H$ through one-step sampling. The generator then updates its parameters by computing the spatial loss $\mathcal{L}_{\text{spatial}}$ in pixel space between the generated image and the ground truth, as well as the adversarial loss $\mathcal{L}_\mathcal{G}$ derived from the discriminator in the latent space (Eq. 2). The loss function for updating the generator is defined as $\mathcal{L}_{\text{spatial}} + \lambda_1 \mathcal{L}_\mathcal{G}$. Specifically, we employ a weighted sum of Mean Squared Error (MSE) loss and perceptual loss to define the spatial loss:

$$
\begin{aligned}
\mathcal{L}_{\text{spatial}}(\mathcal{G}_\theta(x_L), x_H) = \\
\mathcal{L}_{\text{MSE}}(\mathcal{G}_\theta(x_L), x_H) + \lambda_2 \mathcal{L}_{\text{EA-DISTS}}(\mathcal{G}_\theta(x_L), x_H),
\end{aligned}
\tag{8}
$$

where $\lambda_1$ and $\lambda_2$ are hyperparameters used to balance the contributions of each loss component. The supplementary material provide a detailed description of the algorithm's pseudocode and the values of the hyperparameters.

For discriminator training, we utilize paired training features, where each pair consists of a negative sample feature $\hat{z}_H$ and the corresponding real image's latent representation $z_H$ as a positive one. Using Eq. 3, we compute the adversarial loss $\mathcal{L}_\mathcal{D}$ to update the discriminator's parameters. Furthermore, the discriminator can be initialized with weights from more powerful pre-trained models, such as SDXL [19], to achieve superior performance.

This training approach allows our D$^3$SR to overcome the limitations imposed by teacher models, enhancing generator performance without increasing its parameter count or compromising efficiency. Additionally, the integration of a robust discriminator initialized with advanced pre-trained models ensures that the generator receives high-quality feedback, facilitating the production of more realistic and detailed high-resolution images.

Table 1: Quantitative results ($\times 4$) on the Real-ISR testset with ground truth. The best and second-best results are colored red and blue. In the one-step diffusion models, the best metric is **bolded**.

| Dataset | Method | PSNR↑ | SSIM↑ | LPIPS↓ | DISTS↓ | NIQE↓ | MUSIQ↑ | MANIQA↑ | CLIPIQA↑ |
|---|---|---|---|---|---|---|---|---|---|
| **RealSR** | StableSR-s200 | 26.28 | 0.7733 | 0.2622 | 0.1583 | 4.892 | 60.53 | 0.5570 | 0.4310 |
| | DiffBIR-s50 | 24.87 | 0.6486 | 0.3834 | 0.2015 | 3.947 | 68.02 | 0.6309 | 0.6042 |
| | SeeSR-s50 | 26.20 | 0.7555 | 0.2806 | 0.1784 | 4.540 | 66.37 | 0.6118 | 0.5483 |
| | ResShift-s15 | 25.45 | 0.7246 | 0.3727 | 0.2344 | 7.349 | 56.18 | 0.5004 | 0.4307 |
| | ADDSR-s4 | 23.15 | 0.6662 | 0.3769 | 0.2353 | 5.256 | 66.54 | 0.6581 | 0.5390 |
| | SinSR-s1 | 25.83 | 0.7183 | 0.3641 | 0.2193 | 5.746 | 61.62 | 0.5362 | 0.4691 |
| | OSEDiff-s1 | 24.57 | 0.7202 | 0.3036 | 0.1808 | 4.344 | 67.31 | 0.6148 | 0.5524 |
| | ADDSR-s1 | 25.23 | 0.7295 | 0.2990 | 0.1852 | 5.223 | 63.08 | 0.5457 | 0.4498 |
| | **D³SR-s1** | 24.11 | 0.7152 | **0.2961** | **0.1782** | **3.899** | **68.23** | **0.6383** | **0.5647** |
| **DIV2K-val** | StableSR-s200 | 23.68 | 0.6270 | 0.4167 | 0.2023 | 4.602 | 49.51 | 0.4774 | 0.3775 |
| | DiffBIR-s50 | 22.33 | 0.5133 | 0.4681 | 0.1889 | 3.156 | 70.07 | 0.6307 | 0.6352 |
| | SeeSR-s50 | 23.21 | 0.6114 | 0.3477 | 0.1706 | 3.591 | 67.99 | 0.5959 | 0.5842 |
| | ResShift-s15 | 23.55 | 0.6023 | 0.4088 | 0.2228 | 6.870 | 56.07 | 0.4791 | 0.4269 |
| | ADDSR-s4 | 22.08 | 0.5578 | 0.4169 | 0.2145 | 4.738 | 68.26 | 0.5998 | 0.6007 |
| | SinSR-s1 | 22.55 | 0.5405 | 0.4390 | 0.2033 | 5.620 | 62.25 | 0.5011 | 0.5206 |
| | OSEDiff-s1 | 23.10 | 0.6127 | 0.3447 | 0.1750 | 3.583 | 66.62 | 0.5530 | 0.5330 |
| | ADDSR-s1 | 22.74 | 0.6007 | 0.3961 | 0.1974 | 4.270 | 62.08 | 0.5118 | 0.4868 |
| | **D³SR-s1** | 22.05 | 0.6031 | 0.3556 | **0.1500** | **3.295** | **68.51** | **0.5795** | **0.5370** |

## 4 Experiments

We conduct comprehensive experiments to validate the effectiveness of D³SR in real-world image super-resolution (Real-ISR). We provide a detailed introduction of our experimental setup in section 4.1. In section 4.2, we evaluate our method and compare it against the current state-of-the-art methods. In section 4.3, we carry out comprehensive ablation studies to validate the effectiveness and robustness of our proposed approach.

### 4.1 Experimental Settings

**Datasets.** We train D³SR on LSDIR [62] and the first 10k images from FFHQ [63], totaling 95k images. During training, we randomly crop patches of size $512\times512$ pixels from these images. To get low-resolution (LR) and high-resolution (HR) pairs for training, we apply the Real-ESRGAN [64] degradation pipeline. We conduct extensive evaluations of D³SR on a synthetic dataset DIV2K-val [65] and two real-world datasets, including RealSR [66] and RealSet65 [67]. In DIV2K-val, we use the Real-ESRGAN degradation pipeline to synthesize the corresponding LR images. We evaluate our model and all other methods by using the whole images from each dataset.

**Compared Methods.** We compare our D³SR with state-of-the-art DM-based methods for real image super-resolution (Real-ISR), as well as other prominent approaches, including GAN-based and Transformer-based methods. The DM-based methods include multi-step diffusion models, such as StableSR [43], ResShift [67], DiffBIR [16], and SeeSR [14], alongside recently proposed one-step diffusion models like SinSR [30], OSEDiff [32], and AddSR [31]. Other methods include GAN-based approaches, such as BSRGAN [9], RealSR-JPEG [39], Real-ESRGAN [64], LDL [68], and FeMASR [69], as well as Transformer-based method SwinIR [10].

**Evaluation Metrics.** To assess the performance of each method, we employ four full-reference (FR) and four no-reference (NR) image quality metrics. The FR metrics include PSNR, SSIM, LPIPS [60], and DISTS [61]. Both PSNR and SSIM are computed on the Y channel in the YCbCr color space. The NR metrics include NIQE [70], MUSIQ [71], ManIQA [72], and CLIPIQA [73].

**Implementation Details.** We initialize the generator network with the SD 2.1-base parameters and the discriminator network with partial parameters from SDXL. We set both the rank and scaling factor $\alpha$ of LoRA to 16 in the generator and discriminator. We use the AdamW optimizer and set the learning rate for both the generator and discriminator to 5e-5. Training is performed with a batch size of 8 over 100K iterations with 4 NVIDIA A100-40GB GPUs.

### 4.2 Comparison with State-of-the-Art Methods

**Quantitative Results.** Tables 1 and 2 provide quantitative comparisons of the methods across the three datasets. D³SR achieves the best performance among all no-reference (NR) metrics in one-step diffusion methods. These NR metrics reflect the details and realism of the images. This indicates that D³SR outperforms all one-step methods and most multi-step methods. Recently, some studies [18] have pointed out that reference-based metrics such as PSNR and SSIM cannot accurately reflect the

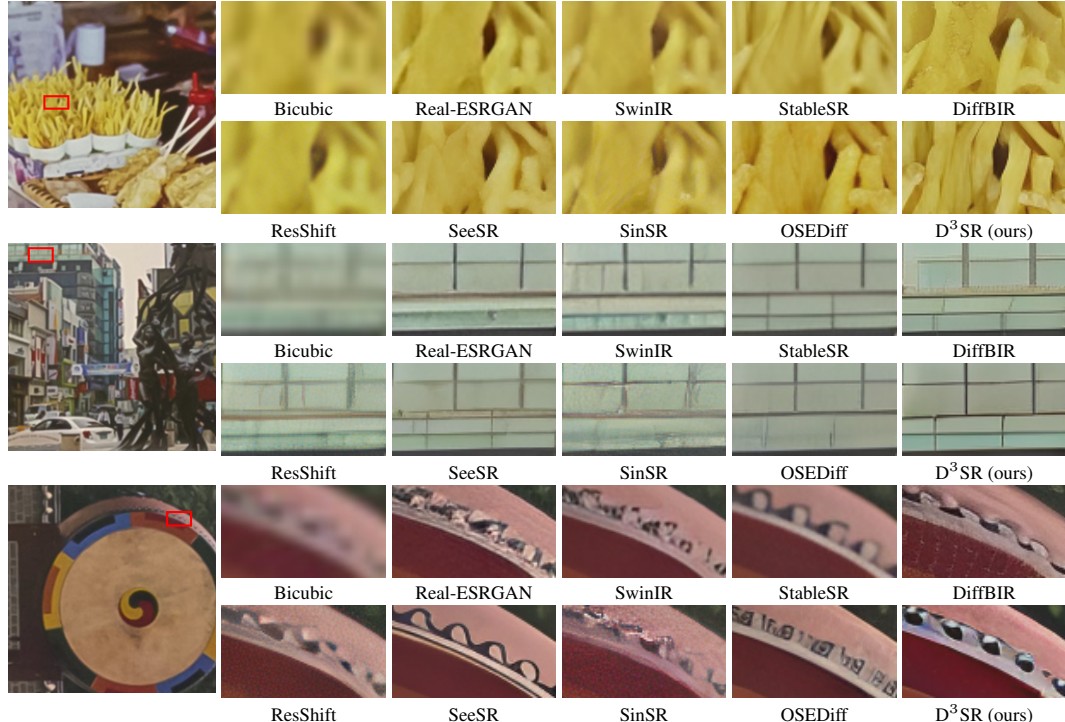

Figure 5: Visual comparisons (×4) on Real-ISR task (RealSR [66] dataset).

performance of diffusion-based Real-ISR methods. We discuss this in the supplementary materials. We compare the quantitative and qualitative results, including those of GAN-based methods. We find that, compared to diffusion-based methods, GAN-based methods generally achieve higher PSNR and SSIM. However, the image quality of these methods is not as high.

**Visual Results.** Figure 5 presents a visual comparison of various diffusion-based Real-ISR methods. As observed, most existing methods struggle to generate realistic details and often produce incorrect content in certain regions of the image due to noise artifacts. Notably, our D³SR demonstrates a significant advantage over others, particularly in the restoration of textual content. Additional visual comparison results are provided in the **supplementary material**.

**Complexity Analysis.** Table 3 presents a complexity comparison of Stable Diffusion (SD)-based Real-ISR methods, including the number

Table 2: Quantitative results (×4) on RealSet65 testset. The best and second-best results are colored red and blue. In the one-step diffusion models, the best metric is **bolded**.

| Method | NIQE↓ | MUSIQ↑ | MANIQA↑ | CLIPIQA↑ |
|---|---|---|---|---|
| StableSR-s200 | 4.985 | 58.89 | 0.5269 | 0.4421 |
| DiffBIR-s50 | 4.122 | 71.23 | 0.6371 | 0.5260 |
| SeeSR-s50 | 4.689 | 69.79 | 0.6018 | 0.5657 |
| ResShift-s15 | 6.730 | 59.36 | 0.5071 | 0.4416 |
| ADDSR-s4 | 5.390 | 68.97 | 0.6075 | 0.5272 |
| SinSR-s1 | 5.664 | 64.22 | 0.5338 | 0.5083 |
| OSEDiff-s1 | 4.224 | 69.04 | 0.6024 | 0.5234 |
| ADDSR-s1 | 5.207 | 64.22 | 0.5258 | 0.4718 |
| **D³SR-s1** | **3.998** | **70.25** | **0.6298** | **0.5481** |

of inference steps, inference time, parameter numbers, and MACs (Multiply-Accumulate Operations). All methods are evaluated on an NVIDIA A100 GPU. D³SR achieves the fastest inference speed among all SD-based methods. Furthermore, our method supports using a fixed text embedding as the generation condition. Therefore, we do not require CLIP and other additional modules (such as DAPE used by OSEDiff and SeeSR, and ControlNet used by DiffBIR) for inference. Our D³SR has the smallest number of model parameters during inference among Stable Diffusion (SD)-based methods, reducing the parameters by 33% compared to OSEDiff.

In this section, we validate the effectiveness of two key components in D³SR. More ablation experiments can be found in the supplementary materials.

### 4.3 Ablation Study
**Perceptual Loss.** Table 4a presents the impact of different perceptual loss functions, as well as only mean squared error (MSE) is applied as the spatial loss. Figure 6 showcases the visual outcomes of these experiments. The results indicate that incorporating perceptual loss is crucial for training SR models, as it facilitates the generation of more realistic details and enhances overall visual quality.

Table 3: Complexity comparison ($\times 4$) among different methods, including sampling steps during inference, inference time, parameter count, and MACs. Inference time and MACs are tested for an output size of $512\times512$ with a single A100-40GB GPU.

| | StableSR | DiffBIR | SeeSR | ResShift | SinSR | OSEDiff | $D^3$SR |
|---|---|---|---|---|---|---|---|
| # Step | 200 | 50 | 50 | 15 | 1 | 1 | 1 |
| Inference Time / s | 11.50 | 7.79 | 5.93 | 0.71 | 0.16 | 0.35 | **0.11** |
| # Total Param / M | $1.4\times10^3$ | $1.6\times10^3$ | $2.0\times10^3$ | 173.8 | 173.8 | $1.4\times10^3$ | **966.3** |
| # MACs / G | 75,812 | 24,528 | 32,336 | 4,903 | 2,059 | 2,269 | **2,132** |

Table 4: Ablation studys on the effects of perceptual losses and different discriminators.

(a) Ablation on different perceptual losses.

| Loss Function | LPIPS↓ | NIQE↓ | MUSIQ↑ | ManIQA↑ |
|---|---|---|---|---|
| MSE | 0.3626 | 4.446 | 65.35 | 0.5457 |
| LPIPS | 0.3190 | 4.123 | 66.41 | 0.6383 |
| EA-LPIPS | 0.3173 | 4.046 | 67.47 | 0.6403 |
| DISTS | 0.3463 | 3.800 | 67.55 | 0.6406 |
| EA-DISTS | 0.3150 | 3.747 | 68.69 | 0.6436 |

(b) Ablation on different discriminators.

| Discriminator | LPIPS↓ | NIQE↓ | MUSIQ↑ | ManIQA↑ |
|---|---|---|---|---|
| None | 0.3862 | 6.962 | 62.36 | 0.5597 |
| CNN | 0.3402 | 6.139 | 64.36 | 0.5666 |
| Diffusion-GAN | 0.3200 | 4.518 | 67.51 | 0.5800 |
| SD 2.1 (ours) | 0.3166 | 3.925 | 68.08 | 0.6198 |
| SDXL (ours) | 0.3150 | 3.747 | 68.69 | 0.6436 |

Figure 6: Visual results ($\times 4$) of DFOSD with different perceptual losses. The left side shows a comparison of the checkerboard. The right one shows content about some numbers, *i.e.*, '24, 26, 28'.

Our proposed edge-aware DISTS (EA-DISTS) achieves the best performance across various image quality metrics and visual assessments. As shown in Fig. 6, EA-DISTS excels in producing highly realistic details, demonstrating its advantage in perceptual quality. This highlights the effectiveness of EA-DISTS in accurately restoring image textures and details, thereby significantly improving the visual quality.

**Diffusion Discriminator.** We evaluate the impact of various discriminator modules on the training of $D^3$SR, including our used diffusion discriminator, vanilla discriminator (CNN), diffusion-GAN [57] style discriminator, and training without any discriminator. The CNN discriminator operates in the pixel space, while other discriminators operate in the latent space. The experimental results are shown in Table 4b. The comparison between the Diffusion-GAN discriminator and the SD 2.1 discriminator indicates that using a pre-trained Stable Diffusion (SD) model as the discriminator outperforms randomly initialized parameters.

Next, we use SDXL 1.0-base as the discriminator while keeping the generator size unchanged to verify the impact of scaling for diffusion discriminators. As shown in the last two rows of Table 4b, the performance of the one-step diffusion model trained with SDXL as the discriminator outperforms the model trained with SD 2.1 as the discriminator, without requiring any modifications to the generator's architecture. This suggests that $D^3$SR can effectively leverage the strengths of more powerful pre-trained models, and enhance the performance of generator without compromising its efficiency. This breaks the upper bound imposed by the muti-step teacher diffusion model, making further performance improvement simpler.

## 5 Conclusion

In this work, we propose $D^3$SR, a One-Step Diffusion model for Real-ISR. Unlike previous methods that use diffusion distillation, our method breaks the limitations of the teacher. We propose the edge-aware DISTS (EA-DISTS) perceptual loss, which enhances the texture realism and visual quality of the generated images. Our adversarial training strategy allows $D^3$SR to outperform multi-step diffusion models in visual quality. Experiments show that $D^3$SR achieves superior performance and improves image realism. This highlights its potential for efficient image restoration.

## Acknowledgments

This work was supported by Shanghai Municipal Science and Technology Major Project (2021SHZDZX0102), the Fundamental Research Funds for the Central Universities, the Special Project on Technological Innovation Application for the 15th National Games and the National Paralympic Games under Grant 2025B01W0005.

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
