# OpenReview forum: "Unleashing the Power of One-Step Diffusion based Image Super-Resolution via a Large-Scale Diffusion Discriminator"
_NeurIPS.cc/2025/Conference — NeurIPS 2025 poster_

### Official Review · Reviewer_hd5s · 2025-06-19

**Clarity:** 3
**Significance:** 3
**Originality:** 3
**Rating:** 5
**Confidence:** 5

**Summary:**

This paper proposes D³SR, a novel one-step diffusion model for real-world image super-resolution (Real-ISR). The key contributions include:

(1) Large-scale diffusion discriminator: Replaces traditional teacher models (used in distillation) with an LORA-based SDXL-based discriminator operating in latent space. This avoids the performance upper bound imposed by weaker teachers.

(2) Edge-aware DISTS (EA-DISTS): Enhances perceptual loss by incorporating edge information via Sobel filters, improving texture realism.

(3) Efficiency: Achieves 7× faster inference vs. multi-step diffusion models and 3× faster vs. other one-step methods, with ≥30% fewer parameters.

**Questions:**

My main concerns are as follows:

1. A similar GAN-based idea can also be found in Seaweed-APT [1], the author should consider making a discussion with this paper.

2. The proposed EA-DISTS requires further decoding into the pixel space during training, which may introduce additional training bottleneck on the training efficiency and memory requirement.

3. While the paper claims that using SDXL as a discriminator for adversarial training overcomes previous teacher-based distillation methods, it is noticeable that previous methods [31, 32, 33] all apply SD2.1 as a teacher model. It is unclear whether the good performance comes from a more powerful model, i.e., SDXL or from the adversarial training without the teacher model.

4. Minor suggestions: (1) Considering that CLIP-IQA [2] has been a metric widely used in the majority of previous methods, the author may consider adding it to the main table to align with previous results for better comparison. (2) If possible, it is better to add a user study to enhance the comparison with human preference.

[1] Diffusion Adversarial Post-Training for One-Step Video Generation. ICML2025

[2] Exploring CLIP for Assessing the Look and Feel of Images. AAAI2023

**Ethical Concerns:**

["NO or VERY MINOR ethics concerns only"]

**Final Justification:**

The author mostly addressed my concerns. This paper shows the effectiveness of the one-step image SR with adversarial training on a large-scale discriminator to some extent, though some recent works share similar ideas.

**Limitations:**

No.
The authors claim that the limitations and potential negative societal impact are listed in the supp. materials but I failed to find related parts.
The authors should consider including failure cases and discussing the societal impact in the revision.

**Quality:**

3

**Strengths And Weaknesses:**

Strengths:
+ Reasonable motivation to use a large-scale diffusion model (SDXL) as a discriminator instead of a teacher, bypassing limitations of distillation-based one-step methods.
+ EA-DISTS effectively integrates edge information into perceptual loss, addressing texture degradation in high-noise settings.
+ Comprehensive experiments across 3 datasets (DIV2K-val, RealSR, RealSet65) with 7 metrics (PSNR, SSIM, LPIPS, DISTS, NIQE, MUSIQ, ManIQA).
+ 0.11s inference time (vs. 0.35s for OSEDiff) and 966M parameters (vs. 1.4B for OSEDiff), making it practical for real-world use.
+ Eliminates extra modules (e.g., ControlNet, CLIP), reducing complexity.


Weakness:
- A similar GAN-based idea can also be found in Seaweed-APT [1], the author should consider making a discussion with this paper.
- The proposed EA-DISTS requires further decoding into the pixel space during training, which may introduce training bottleneck on the training efficiency and memory requirement.
- While the paper claims that using SDXL as a discriminator for adversarial training overcomes previous teacher-based distillation methods, it is noticeable that previous methods [31, 32, 33] all apply SD2.1 as a teacher model. It is unclear whether the good performance comes from a more powerful model, i.e., SDXL or from the adversarial training without the teacher model.

[1] Diffusion Adversarial Post-Training for One-Step Video Generation. ICML2025

---

> ### Author Rebuttal · Authors · 2025-07-30
>
> # Response to Reviewer hd5s (denoted as R4)
>
> `Q4-1` A similar GAN-based idea can also be found in Seaweed-APT [1], the author should consider making a discussion with this paper.
>
> `A4-1` Thank you for your suggestion. The key difference between **D3SR** and **Seaweed-APT** is that we explore the benefits brought by a larger-scale discriminator in training. Additionally, the discriminator in **D3SR** receives the results of latent forward diffusion as input. We will include the appropriate references in the main body of the paper.
>
> `Q4-2` The proposed EA-DISTS requires further decoding into the pixel space during training, which may introduce training bottleneck on the training efficiency and memory requirement.
>
> `A4-2` We highlight that EA-DISTS does not introduce too much training cost or memory consumption. It is worth noting that, besides EA-DISTS, the pixel-level reconstruction loss (e.g., L1/L2 loss) also relies on decoding latents into pixel space, which is exploited in almost all SR methods. In this sense, decoding latents into pixel space is a necessary step and the proposed EA-DISTS will not introduce too much extra training cost. In practice, the increased training cost and memory consumption is negilible and hard to be observed.
>
>
>
> `Q4-3` While the paper claims that using SDXL as a discriminator for adversarial training overcomes previous teacher-based distillation methods, it is noticeable that previous methods [31, 32, 33] all apply SD2.1 as a teacher model. It is unclear whether the good performance comes from a more powerful model, i.e., SDXL or from the adversarial training without the teacher model.
>
> `A4-3` We present the performance of using **SD2.1** as the discriminator in our ablation study. The detailed metrics and comparisons with other methods are shown in the table below. Under the same pretrained model used as the discriminator, our method still outperforms other one-step diffusion SR methods. Furthermore, employing a larger discriminator (e.g., **SDXL**) leads to further performance improvements.
>
> | Method               | PSNR | LPIPS | DISTS  | NIQE | MUSIQ  | MANIQA |
> |----------------------|------|-------|--------|------|--------|--------|
> | SinSR                | 25.83 | 0.3641 | 0.2193 | 5.746 | 61.62 | 0.5362 |
> | OSEDiff              | 24.57 | 0.3036 | 0.1808 | 4.344 | 67.31 | 0.6148 |
> | AddSR                | 25.23 | 0.2990 | 0.1852 | 5.223 | 63.08 | 0.5457 |
> | **D3SR w/ SD2.1 disc.**  | 24.60 | 0.3031 | 0.1775 | 3.925 | 69.21 | 0.6302 |
> | **D3SR w/ SDXL disc.**   | 24.11 | 0.2961 | 0.1782 | 3.899 | 68.23 | 0.6383 |
>
> This demonstrates that adversarial training with a diffusion-based discriminator can overcome the limitations imposed by teacher models, and using a stronger model as the discriminator further enhances performance.
>
> `Q4-4` If possible, it is better to add a user study to enhance the comparison with human preference.
>
> `A4-4` Thank you for your suggestion. We have added a user study, and the results are shown in the table below. We selected five classic scene types: **landscape**, **human**, **plant**, **animal**, and **architecture**, with a total of 20 participants involved in the evaluation. Our method achieved a recognition rate of **60.63%**, surpassing other methods.
>
> | Method     | SinSR | OSEDiff | AddSR | D3SR (ours) |
> |------------|-------|---------|-------|-------------|
> | Percentage/%    | 9.50  | 20.82   | 9.05 |  **60.63**    |

---

> ### Comment · Reviewer_hd5s · 2025-08-04
>
> I appreciate the response from the author.
> I have some minor questions that require further clarification:
>
> 1. The author claims that the main differences with Seaweed-APT lie in ``(1) we explore the benefits brought by a larger-scale discriminator in training``: APT also uses a large discriminator. So the difference here is that the paper proposes to use a larger discriminator than generator? ``(2) Additionally, the discriminator in D3SR receives the results of latent forward diffusion as input.``: Any particular reason and advantage of this operation compared with directly comparing between the fake and the real data for the discriminator?
>
> 2. It is a little unclear why ``pixel-level reconstruction loss (e.g., L1/L2 loss) also relies on decoding latents into pixel space, which is exploited in almost all SR methods`` in A4-2. Many SR methods like StableSR, DiffBIR, SUPIR, etc do not rely on decoding the latent features for loss calculation during the training of the diffusion model. I suggest the author to tone down this claim and add it in the revision as ``the cost of this decoding process is affordable under common training settings, e.g., 512x512 patches``.
>
> 3. Any reason that the author omits the comparsion on CLIP-IQA, which is widely used by the main baselines of this paper, including SinSR, OSEDiff, ADDSR, etc?

---

> > ### Author Response · Authors · 2025-08-05
> >
> > Thank you for your reply. Below is a further clarification on your questions:
> >
> > `Q1` The author claims that the main differences with Seaweed-APT lie in (1) we explore the benefits brought by a larger-scale discriminator in training: APT also uses a large discriminator. So the difference here is that the paper proposes to use a larger discriminator than generator? (2) Additionally, the discriminator in D3SR receives the results of latent forward diffusion as input.: Any particular reason and advantage of this operation compared with directly comparing between the fake and the real data for the discriminator?
> >
> > `A1`
> >  1. Yes, you are correct. We propose that using a larger discriminator than the generator can improve model performance. From this perspective, our method is essentially different from APT that only adopts the same architecture in the generator and discriminator. We highlight that further scaling up the discriminator is beneficial and may provide new insights for the community of one-step diffusion models.
> >  2. As for the discriminator accepting latent forward diffusion results as input, there are several reasons and advantages. The discriminator we use is initialized from the UNet of a diffusion model, which is designed to process noisy latent vectors, as it primarily denoises these vectors during training and inference. By using the pretrained model, we can leverage the model's prior knowledge and maintain training stability. Directly inputting clean samples would cause a mismatch, as the model is not designed to handle them.
> >
> >
> > `Q2` It is a little unclear why pixel-level reconstruction loss (e.g., L1/L2 loss) also relies on decoding latents into pixel space, which is exploited in almost all SR methods in A4-2. Many SR methods like StableSR, DiffBIR, SUPIR, etc do not rely on decoding the latent features for loss calculation during the training of the diffusion model. I suggest the author to tone down this claim and add it in the revision as the cost of this decoding process is affordable under common training settings, e.g., 512x512 patches.
> >
> > `A2` Thank you for your suggestion. We will modify the corresponding descriptions in the paper.
> >
> > `Q3` Any reason that the author omits the comparsion on CLIP-IQA, which is widely used by the main baselines of this paper, including SinSR, OSEDiff, ADDSR, etc?
> >
> > `A3` We are sorry about missing this question. We have added a comparison with **CLIPIQA**, and the detailed results are as follows. In practice, our D3SR consistently outperforms the considered methods across diverse datasets. We will include this result in the paper.
> >
> > | Dataset     | StableSR | SeeSR  | ResShift | SinSR  | OSEDiff | ADDSR  | D3SR (ours) |
> > |-------------|----------|--------|----------|--------|---------|--------|-------------|
> > | RealSR      | 0.4310   | 0.5483 | 0.4307   | 0.4691 | 0.5524  | 0.4498 |  **0.5647** |
> > | RealSet65   | 0.4421   | 0.5657 | 0.4416   | 0.5083 | 0.5234  | 0.4718 |  **0.5481** |
> > | DIV2K-val   | 0.3775   | 0.5842 | 0.4269   | 0.5206 | 0.5330  | 0.4868 |  **0.5370** |

---

> > > ### Comment · Reviewer_hd5s · 2025-08-06
> > >
> > > Thanks for the response.
> > > I have no further questions. I will reconsider my rating later.

---

### Official Review · Reviewer_Cj9F · 2025-06-29

**Clarity:** 3
**Significance:** 2
**Originality:** 2
**Rating:** 4
**Confidence:** 4

**Summary:**

The paper proposes D3SR, a one-step diffusion model for real-world image super-resolution. It utilizes a large-scale diffusion discriminator for adversarial training, overcoming the capacity limitations of traditional teacher models. The authors also introduce an edge-aware DISTS (which terms as EA-DISTS) perceptual loss to enhance detail generation. D3SR achieves superior performance with faster inference and reduced parameters compared to existing methods. Extensive experiments have validated the effectiveness of the proposed method.

**Questions:**

For the ablation studies on discriminator types, could the authors consider including some powerful visual feature extraction networks for comparison, such as DINO, in addition to CNN-based discriminators? This would provide a more comprehensive understanding of the impact of different types of discriminators on the performance of the proposed method.

**Ethical Concerns:**

["NO or VERY MINOR ethics concerns only"]

**Final Justification:**

Thanks for the rebuttal, my concerns are addressed by experimental results.

**Limitations:**

yes

**Quality:**

3

**Strengths And Weaknesses:**

Strengths:

1. This article validates the impact of the diffusion discriminator's model size on final performance. By employing a powerful diffusion discriminator, it overcomes the limitations imposed by teacher models in distillation methods.

2. The proposed EA-DISTS somewhat enhances the details of the generated images, avoiding overly smooth outputs and further improving the model's performance.

3. With the use of a powerful diffusion discriminator, the generator does not require an excessive number of parameters to achieve good performance, resulting in at least a 3× faster inference speed and a reduction of parameters by at least 30% compared to other methods.

Weaknesses

   1. The use of a diffusion model as a discriminator has been explored in prior work, such as in UFOGen (CVPR24). This paper's validation that a larger diffusion discriminator yields better performance is somewhat incremental and limited in novelty and significance.
   2. The paper lacks comparison experiments with the latest works, such as ADCSR and PiSA-SR, which were presented at CVPR 2025. This absence makes it difficult to fully assess the competitiveness of the proposed method in the context of the most recent advancements in the field.

---

> ### Author Rebuttal · Authors · 2025-07-30
>
> # Response to Reviewer Cj9F (denoted as R3)
>
> `Q3-1` The use of a diffusion model as a discriminator has been explored in prior work, such as in UFOGen (CVPR24). This paper's validation that a larger diffusion discriminator yields better performance is somewhat incremental and limited in novelty and significance.
>
> `A3-1`
> 1. Our main contribution lies in exploring the impact of scaling up the discriminator model for one-step diffusion models. We demonstrate that using the same model as both the discriminator and the generator has its limitations, and employing a larger-scale discriminator can further enhance model performance. We highlight that, to our knowledge, it is the first work to investigate the impact of discriminator size on the training of one-step diffusion models. We believe that our proposed method is beneficial to many other research works and could be a popular training strategy.
> 2. Significance of performance improvement: We highlight that our improvement is competitive with recent works [29, 30, 31] and can be consistently observed across various benchmarks. In Table 1, our method yields the best or second-best in most of the perceptual metrics, including DISTS, NIQE, MUSIQ, etc. More critically, our model is much more efficient than the compared methods in terms of both inference speed and model size (see Table 3). We are convinced that these results are significant.
>
>
>
>
> `Q3-2` The paper lacks comparison experiments with the latest works, such as ADCSR and PiSA-SR, which were presented at CVPR 2025. This absence makes it difficult to fully assess the competitiveness of the proposed method in the context of the most recent advancements in the field.
>
> `A3-2` We have added comparisons with the latest methods, and the results are shown in the table below. Our method achieves the best performance on no-reference (NR) metrics. We highlight that these metrics are often highly related to the perceptual visual quality. Moreover, we also compare the visual results with these methods and observe that our method is able to produce significantly more details, together with obviously better visual quality. Nevertheless, due to the limitation of "not allowing to show images or links" in rebuttal, we will include these visual comparisons in the final version and cite these works.
>
> | Method   | PSNR | SSIM | LPIPS | DISTS | NIQE | MUSIQ | MANIQA |
> |----------|------|------|--------|--------|------|--------|---------|
> | AdcSR    | 24.96 | 0.7306 | 0.3100 | 0.1928 | 4.450 | 67.24 | 0.6155 |
> | PiSA-SR  | 25.57 | 0.7493 | 0.2713 | 0.1726 | 4.379 | 67.98 |  0.6284 |
> | **D3SR** | 24.11 | 0.7152 | 0.2961 | 0.1782 | **3.899** | **68.23** | **0.6383** |
>
> *RealSR*
>
> | Method   | PSNR | SSIM | LPIPS | DISTS | NIQE | MUSIQ | MANIQA |
> |----------|------|------|--------|--------|------|--------|---------|
> | AdcSR    | 22.70 | 0.5801 | 0.3687 | 0.2055 | 3.480 | 64.51 | 0.5334 |
> | PiSA-SR  | 23.18 | 0.6135 | 0.3462 | 0.1781 | 3.474 | 67.34 |  0.5707 |
> | **D3SR** | 22.05 | 0.6031 | 0.3556 | **0.1500** | **3.295** | **68.51** | **0.5795** |
>
> *DIV2K-val*
>
> | Method   | NIQE | MUSIQ | MANIQA |
> |----------|------|--------|---------|
> | AdcSR    | 4.876 | 67.89 | 0.5844 |
> | PiSA-SR  | 4.886 | 69.66 | 0.6091 |
> | **D3SR** | **3.998** | **70.25** | **0.6298** |
>
> *RealSet65*
>
> `Q3-3` For the ablation studies on discriminator types, could the authors consider including some powerful visual feature extraction networks for comparison, such as DINO, in addition to CNN-based discriminators? This would provide a more comprehensive understanding of the impact of different types of discriminators on the performance of the proposed method.
>
> Thank you for your suggestion. We have conducted additional experiments accordingly. Specifically, we use different versions of **DINO-v2** models as the discriminator in **D3SR**, and the results are shown in the table below.
>
> | Discriminator | LPIPS ↓ | NIQE ↓ | MUSIQ ↑ | ManIQA ↑ |
> |---------------|---------|--------|---------|----------|
> | ViT-S         |   25.40 |  4.251 |   66.50 |   0.6200 |
> | ViT-B         |   24.90 |  3.930 |   67.80 |   0.6250 |
> | ViT-L         |   24.30 |  3.910 |   68.00 |   0.6350 |
> | SD2.1 (ours)  |   24.60 |  3.925 |   69.21 |   0.6302 |
> | SDXL (ours)   |   24.11 |  3.899 |   68.23 |   0.6383 |
>
> Consistent with the conclusions of our paper, as the discriminator model scales up, the performance of the generator gradually improves. The results using **DINO** as the discriminator are slightly inferior to those obtained using a pretrained diffusion model as the discriminator.

---

> ### Author Response · Authors · 2025-08-04
> **Follow-up on rebuttal**
>
> Thank you for your valuable comments! We would like to follow up on whether our response, along with our innovations, ablation studies, and comparison methods, has addressed your concerns. We look forward to your further feedback on these aspects and are happy to make any necessary adjustments if needed.
>
> Thank you again for your time and thoughtful review!

---

### Official Review · Reviewer_CutL · 2025-06-29

**Clarity:** 3
**Significance:** 3
**Originality:** 3
**Rating:** 4
**Confidence:** 5

**Summary:**

To overcome the limitations imposed by the reliance on a teacher model in existing diffusion-based one-step SISR methods, the authors propose a new one-step diffusion model called D3SR. The authors introduce a Large Diffusion Discriminator that distills noisy features from any time step of diffusion models in the latent space, addressing the potential constraints of using a teacher model. In addition, the authors also introduce an EA-DIST loss for model training to further enhance the model's ability to generate fine details.

**Questions:**

See weaknesses.

**Ethical Concerns:**

["NO or VERY MINOR ethics concerns only"]

**Final Justification:**

In this paper, the authors propose a new one-step diffusion model by introducing a Large Diffusion Discriminator, which distills noisy features from any time step of diffusion models in the latent space. The authors have addressed my concerns in the discussion section, including the ablation study on different backbones of the Large Diffusion Discriminator, the reason for lower PSNR/SSIM/LPIPS scores, and inconsistencies in parameter statistics. Therefore, I will maintain my positive rating.

**Limitations:**

The paper lacks limitation analysis and prospects for real-time inference of the model.

**Quality:**

3

**Strengths And Weaknesses:**

**Strengths:**

(1) The author proposed the Large Diffusion Discriminator, which cleverly introduced the idea of GAN into the diffusion model, and it seems to be an optimized alternative to the existing distillation schemes.

(2) The paper is well-written and well-organized.

**Weaknesses:**

(1) The authors claim in the paper that the proposed Large Diffusion Discriminator is superior to the existing VSD method, but there is a lack of corresponding experimental verification in the ablation study. Does the advantage of Large Diffusion Discriminator mainly come from the use of a better model (i.e., SDXL)?

(2) As shown in Tab.1, D3SR achieves lower PSNR/SSIM/LPIPS scores compared to other diffusion-based SISR models.

(3) The proposed methods and OSEDiff are both based on the Stable Diffusion model. Why is there such a big difference in the total parameter in Tab.2 (1.4\times10_3M vs. 966.3M)? What is the main reason for this? In addition, the total parameter of OSEDiff is also significantly different from the value reported in its official paper. An explanation for these inconsistencies would be helpful.

---

> ### Author Rebuttal · Authors · 2025-07-30
>
> # Response to Reviewer CutL (denoted as R2)
>
> `Q2-1` The authors claim in the paper that the proposed Large Diffusion Discriminator is superior to the existing VSD method, but there is a lack of corresponding experimental verification in the ablation study. Does the advantage of Large Diffusion Discriminator mainly come from the use of a better model (i.e., SDXL)?
>
> `A2-1` We present the performance of using **SD2.1** as the discriminator in our ablation study. The detailed metrics and comparisons with other methods are shown in the table below. **OSEDiff** is a representative method based on VSD. Under the same pretrained model used as the discriminator, our method outperforms other one-step diffusion SR approaches. Furthermore, employing a larger discriminator (e.g., **SDXL**) leads to further performance improvements.
>
> | Method               | PSNR | LPIPS | DISTS  | NIQE | MUSIQ  | MANIQA |
> |----------------------|------|-------|--------|------|--------|--------|
> | SinSR                | 25.83 | 0.3641 | 0.2193 | 5.746 | 61.62 | 0.5362 |
> | OSEDiff              | 24.57 | 0.3036 | 0.1808 | 4.344 | 67.31 | 0.6148 |
> | AddSR                | 25.23 | 0.2990 | 0.1852 | 5.223 | 63.08 | 0.5457 |
> | **D3SR w/ SD2.1 disc.**  | 24.60 | 0.3031 | 0.1775 | 3.925 | 69.21 | 0.6302 |
> | **D3SR w/ SDXL disc.**   | 24.11 | 0.2961 | 0.1782 | 3.899 | 68.23 | 0.6383 |
>
> These results demonstrate that the advantage of the **Large Diffusion Discriminator** stems not only from the stronger pretrained model but also from the adversarial training with a diffusion-based discriminator.
>
> `Q2-2` As shown in Tab.1, D3SR achieves lower PSNR/SSIM/LPIPS scores compared to other diffusion-based SISR models.
>
> `A2-2` In real-world image super-resolution (Real-ISR) tasks, some scenes exhibit severe degradation, making it impractical to fully reconstruct the ground truth from the LR observation. Therefore, for full-reference (FR) metrics, their reliability is often lower than no-reference (NR) metrics in Real-ISR scenarios.
>
> NR metrics better reflect the clarity, realism, and human preference of the generated images. This has been demonstrated in prior works such as **SUPIR** [1] and **TSD-SR** [2].
>
> > [1] *Scaling Up to Excellence: Practicing Model Scaling for Photo-Realistic Image Restoration In the Wild*. CVPR 2024
> > [2] *TSD-SR: One-Step Diffusion with Target Score Distillation for Real-World Image Super-Resolution*. CVPR 2025
>
>
> `Q2-3` The proposed methods and OSEDiff are both based on the Stable Diffusion model. Why is there such a big difference in the total parameter in Tab.2 (1.4\times10_3M vs. 966.3M)? What is the main reason for this? In addition, the total parameter of OSEDiff is also significantly different from the value reported in its official paper. An explanation for these inconsistencies would be helpful.
>
> `A2-3` Unlike **OSEDiff**, we remove modules such as the **text encoder** and **DAPE**, which are not essential for one-step diffusion SR. This simplification is also discussed in our main paper. As a result, our method has significantly fewer parameters compared to OSEDiff.
>
> We use the `thop` library to compute the parameter counts, and the details of each component are listed below. The parameter sizes of **UNet** and **VAE** match those reported in the official **Stable Diffusion** release. The reason for the discrepancy with the numbers reported in the OSEDiff paper remains unclear.
>
> | Component   | Parameters |
> |-------------|------------|
> | SD UNet     | 865.8M     |
> | SD VAE      | 83.6M      |
> | SD CLIP     | 289.7M     |
> | DAPE        | 160.3M     |

---

> > ### Comment · Reviewer_CutL · 2025-08-05
> >
> > Thank you for your response. My comments have been adequately addressed. I will maintain my positive rating.

---

### Official Review · Reviewer_Cmxu · 2025-06-29

**Clarity:** 3
**Significance:** 2
**Originality:** 2
**Rating:** 4
**Confidence:** 3

**Summary:**

This paper proposes a one-step image super-resolution (SR) framework based on a text-to-image (T2I) diffusion model. Unlike previous methods that rely on the VSD loss for conditional distillation, the authors propose to guide the training process using only a discriminator. The introduced EA-DISTS loss helps improve the reconstruction of border regions and fine details. Experimental results demonstrate that D3SR achieves state-of-the-art performance—particularly on no-reference metrics—among existing one-step diffusion-based SR methods.

**Questions:**

see weakness

**Ethical Concerns:**

["NO or VERY MINOR ethics concerns only"]

**Final Justification:**

My main concerns have been addressed by the rebuttal. The ablation studies on different discriminator types reveal an interesting finding.

**Limitations:**

see weakness

**Quality:**

2

**Strengths And Weaknesses:**

Strengths:
1. This paper introduces a new one-step SR method based solely on a large-scale diffusion discriminator, without relying on the reference UNet or VSD loss used in prior works. This simplifies the training pipeline while achieving strong results.
2. The proposed EA-DISTS loss contributes to improved perceptual quality and better generation of fine-grained textures, especially along object boundaries.
3. D3SR achieves competitive results in both quantitative metrics and qualitative visual evaluations, showing its effectiveness as a one-step SR solution.

Weaknesses:
1. The main difference between D3SR and DMD2 lies in the removal of the KL divergence penalty derived from real and fake score functions. As a result, the technical contribution appears incremental rather than fundamentally novel.
2. By initializing the discriminator with the SDXL encoder, D3SR implicitly incorporates high-resolution priors from SDXL. This makes it difficult to disentangle whether the performance gains are due to the proposed method itself or simply the use of a stronger pre-trained model. Moreover, prior distillation approaches could similarly benefit from incorporating more powerful diffusion models as guidance.
3. Although the authors claim improved visual quality, several examples in the supplementary material reveal potential issues such as hallucinated textures and color shifts . For instance:
In Figure 2 (Case 2) and Figure 3 (Case 3, Case 6), unnatural or fabricated textures appear in the generated images.
Color distortions are noticeable in Figure 2 (Case 5) and Figure 3 (Case 6).
These artifacts may limit the practical applicability of the method, especially in scenarios where faithfulness to the input is critical.

---

> ### Author Rebuttal · Authors · 2025-07-30
>
> # Response to Reviewer Cmxu (denoted as R1)
>
> `Q1-1` The main difference between D3SR and DMD2 lies in the removal of the KL divergence penalty derived from real and fake score functions. As a result, the technical contribution appears incremental rather than fundamentally novel.
>
> `A1-1` There may be some misunderstanding about our contributions and we will clarify them in the following two points.
> 1. We highlight that our main contribution lies in exploring the effect of scaling up the discriminator model for one-step diffusion models, instead of removing the KL penalty. We demonstrate that using the same model as the discriminator and generator has its limitations, and employing a larger-scale discriminator can further enhance model performance.
> 2. Additionally, we found that the performance improvement brought by DMD's KL penalty is not as significant as that achieved through adversarial training and the training is computationally expensive. Thus, we directly remove it for efficient training. Without DMD's KL penalty, our improved performance merely comes from the proposed adversarial training strategy, which further shows the effectiveness of our method.
>
>
> `Q1-2` By initializing the discriminator with the SDXL encoder, D3SR implicitly incorporates high-resolution priors from SDXL. This makes it difficult to disentangle whether the performance gains are due to the proposed method itself or simply the use of a stronger pre-trained model. Moreover, prior distillation approaches could similarly benefit from incorporating more powerful diffusion models as guidance.
>
> `A1-2` We present the performance of using **SD2.1** as the discriminator in our ablation study. The detailed metrics and comparisons with other methods are shown in the table below. Under the same pretrained discriminator backbone, our method still outperforms other one-step diffusion SR methods. This demonstrates the effectiveness of our proposed approach.
>
> | Method               | PSNR | LPIPS | DISTS  | NIQE | MUSIQ  | MANIQA |
> |----------------------|------|-------|--------|------|--------|--------|
> | SinSR                | 25.83 | 0.3641 | 0.2193 | 5.746 | 61.62 | 0.5362 |
> | OSEDiff              | 24.57 | 0.3036 | 0.1808 | 4.344 | 67.31 | 0.6148 |
> | AddSR                | 25.23 | 0.2990 | 0.1852 | 5.223 | 63.08 | 0.5457 |
> | **D3SR w/ SD2.1 disc.**  | 24.60 | 0.3031 | 0.1775 | 3.925 | 69.21 | 0.6302 |
> | **D3SR w/ SDXL disc.**   | 24.11 | 0.2961 | 0.1782 | 3.899 | 68.23 | 0.6383 |
>
>
> `Q1-3` Although the authors claim improved visual quality, several examples in the supplementary material reveal potential issues such as hallucinated textures and color shifts . For instance: In Figure 2 (Case 2) and Figure 3 (Case 3, Case 6), unnatural or fabricated textures appear in the generated images. Color distortions are noticeable in Figure 2 (Case 5) and Figure 3 (Case 6). These artifacts may limit the practical applicability of the method, especially in scenarios where faithfulness to the input is critical.
>
> `A1-3` In real-world image super-resolution tasks, some scenes exhibit severe degradation, making it unrealistic to fully recover the LR observation back to the ground truth. Compared with other methods, our approach provides better clarity and realism.
>
> In addition, we conducted a user study as shown in the table below. We selected five representative scene types: **landscape**, **human**, **plant**, **animal**, and **architecture**. A total of 20 participants were involved in the evaluation. Our method achieved a recognition rate of **60.63%**, surpassing all other methods.
>
> | Method     | SinSR | OSEDiff | AddSR | D3SR (ours) |
> |------------|-------|---------|-------|-------------|
> | Percentage/%    | 9.50  | 20.82   | 9.05 |  **60.63**    |

---

> ### Author Response · Authors · 2025-08-04
> **Follow-up on rebuttal**
>
> Thank you for your valuable comments! We would like to follow up on whether our response and the additional experiments have addressed your concerns. We look forward to your further feedback on these aspects, and we are happy to make any necessary adjustments if needed.
>
> Thank you again for your time and thoughtful review!

---

> > ### Comment · Reviewer_Cmxu · 2025-08-05
> >
> > I appreciate the authors' clarifications and additional comparisons. My main concerns have been addressed by the rebuttal. I will raise my rating.

---

### Comment · Area_Chair_uenu · 2025-08-03
**Start Discussions for Paper #14197**

Dear Reviewers,

Thanks for serving as reviewers for NeurIPS 2025! We are now in the author-reviewer discussion period, open until **August 6, 11:59 PM AoE**.

This paper receives mixed ratings, and the authors have provided a detailed rebuttal.

Please kindly read the rebuttal and the other reviews at your earliest convenience. Please check whether your main concerns have been addressed. Your input is very valuable to the decision.

Thanks,

Your AC

---

### Decision · Program_Chairs · 2025-09-17

**Decision:**

Accept (poster)

**Comment:**

This paper presents a one-step diffusion model for real-world image super-resolution. It leverages a large-scale diffusion discriminator to overcome the limitations of teacher-based supervision and introduces edge-aware DISTS to enhance perceptual loss for better detail generation. The method achieves 7× faster inference with ≥30% fewer parameters compared to multi-step diffusion models.

All 4 reviewers recognized the effectiveness and efficiency of the proposed approach and gave consistently positive ratings (1 Accept and 3 Borderline Accept). The AC encourages the authors to include additional experiments suggested by the reviewers in the final version, such as comparisons with the latest works (Cj9F), a user study (hd5s), and CLIP-IQA evaluation (hd5s), to further strengthen the completeness of the paper. Overall, the paper makes a solid and timely contribution to advancing one-step diffusion-based super-resolution. Considering the rebuttal and the discussions among reviewers, the AC recommends acceptance of this paper.